# My Caregiver the Cobot: Comparing Visualization Techniques to Effectively Communicate Cobot Perception to People with Physical Impairments

**DOI:** 10.3390/s22030755

**Published:** 2022-01-19

**Authors:** Max Pascher, Kirill Kronhardt, Til Franzen, Uwe Gruenefeld, Stefan Schneegass, Jens Gerken

**Affiliations:** 1Human-Computer Interaction Group, Department of Media Informatics and Communication, Westphalian University of Applied Sciences, 45897 Gelsenkirchen, Germany; kirill.kronhardt@studmail.w-hs.de (K.K.); til.franzen@studmail.w-hs.de (T.F.); jens.gerken@w-hs.de (J.G.); 2Human-Computer Interaction Group, Paluno—The Ruhr Institute for Software Technology, Faculty of Business Administration and Economics, University of Duisburg-Essen, 45127 Essen, Germany; uwe.gruenefeld@uni-due.de (U.G.); stefan.schneegass@uni-due.de (S.S.)

**Keywords:** cobot, human–robot collaboration, visualization techniques, projection, virtual reality

## Abstract

Nowadays, robots are found in a growing number of areas where they collaborate closely with humans. Enabled by lightweight materials and safety sensors, these cobots are gaining increasing popularity in domestic care, where they support people with physical impairments in their everyday lives. However, when cobots perform actions autonomously, it remains challenging for human collaborators to understand and predict their behavior, which is crucial for achieving trust and user acceptance. One significant aspect of predicting cobot behavior is understanding their perception and comprehending how they “see” the world. To tackle this challenge, we compared three different visualization techniques for Spatial Augmented Reality. All of these communicate cobot perception by visually indicating which objects in the cobot’s surrounding have been identified by their sensors. We compared the well-established visualizations *Wedge* and *Halo* against our proposed visualization *Line* in a remote user experiment with participants suffering from physical impairments. In a second remote experiment, we validated these findings with a broader non-specific user base. Our findings show that *Line*, a lower complexity visualization, results in significantly faster reaction times compared to *Halo*, and lower task load compared to both *Wedge* and *Halo*. Overall, users prefer *Line* as a more straightforward visualization. In Spatial Augmented Reality, with its known disadvantage of limited projection area size, established off-screen visualizations are not effective in communicating cobot perception and *Line* presents an easy-to-understand alternative.

## 1. Introduction

While robots were previously taught to perform simple repetitive tasks, they have started to evolve into collaborators in our professional and personal lives [1,2]. As a result, these so-called cobots support humans in various ways that were unimaginable just a few years ago. One area that has seen drastic advances in human–robot collaboration is domestic care, with cobots supporting people with physical impairments [3]. These assist people in various ways [4], from activities of daily living (ADLs), including basic tasks such as drinking, eating, and grooming, to leisure-time activities [5,6]. In domestic care, cobots reduce the need for the constant presence of caregivers, empowering people previously reliant on others for help to regain their independence. Our previous research on the needs of people with physical impairments showed a strong desire for privacy and alone time, which can undoubtedly be achieved with reliable robotic support [7].

However, new challenges arise when cobots are tasked with autonomous or semi-autonomous actions, resulting in additional stress for end-users [8]. Close proximity collaboration between humans and cobots remains particularly challenging [9]. These challenges include effective communication to the end-user of (a) motion intent and (b) the spatial perception of the cobot’s vicinity [10]. Accurate communication increases our understanding of the cobot while avoiding the unpredictability regarding impending steps, motions, and sensed environment parameters. While visualizations of motion intent have been extensively studied [9,10,11,12,13,14], communicating cobot perception has received less attention [15]. We define cobot perception as the sensory information acquired to computationally understand the surroundings, including the detection and identification of objects of interest in the physical vicinity. In our work, we communicate these sensory information acquired by the cobot using three different visualization techniques. Users benefit from receiving information about and understanding a cobot’s spatial perception as perception failures, including errors in computer vision and object perception, can occur. Without communication, these are otherwise difficult to predict and to understand [16,17]. Accordingly, there is a clear need to accurately express cobot perception to their human collaborators to improve the correct prediction of the cobot behavior [18].

Augmented reality (AR) technology is a promising medium to communicate cobot perception, with the possibility to directly show relevant perceptual information in the user’s line-of-sight whilst linking 3D with the physical world. In previous work, AR technology has shown encouraging results for the visualization of motion intent [9,12]. Any visualization technique aiding users in understanding the cobot’s perception of its surroundings needs to effectively communicate all objects both within the visible area of the user and outside (or “off-screen”). The off-screen area is defined by the field of view of the user but, more importantly, limited by the means of the AR systems spatial visualization capabilities.

The release of the first Microsoft HoloLens resulted in an increased focus on approaches relying on Head-Mounted Displays (HMDs) [9]. However, even state-of-the-art HMD-AR such as the Microsoft HoloLens 2, (https://www.microsoft.com/de-de/hololens, last retrieved 30 December 2021) have a restricted display area which limits the field of view of the user [19]. Recent studies on the design preferences of people living with physical impairments also revealed that these displays are often impractical or not usable at all for the target population [20]. In addition, HMDs prevent direct information exchange with secondary users such as caregivers, thereby excluding them from providing necessary support. Similarly to HMDs, approaches using Mobile-AR (MAR) also limit the field of view through their display size and orientation, rendering them potentially unusable for people with physical impairments [21].

Spatial Augmented Reality (SAR) is another approach using projection techniques to augment the surface in the environment [22]. While essentially limited to 2D, research for motion intent has shown that SAR can be adapted to cover a dynamic workspace that encompasses multiple surface areas [10,11], e.g., in our scenario, this refers to interacting with objects “on a table” and “retrieving objects from shelves”. Due to the significant decreases in the cost of projection technology and advances in pico-sized projectors, SAR has garnered increased interest in recent years [23]. SAR can augment larger areas of the surroundings, exceeding even the physical field of view of the user, and unlike HMDs, can be observed by secondary users. However, the possible field of view depends on the mounting position of the projection technique. While SAR may increase the visible augmentation area, the problem for effectively communicating off-screen objects still exists and is currently unsolved.

We investigated the potential visualization approaches that communicate the cobot’s perception and particularly the information about detected objects in its physical surroundings. Information about physical objects are critical, as any breakdown in the successful detection of such objects by a (semi-) autonomous cobot can result in errors in behavior with the potential to harm the user, such as knocking over objects or even destroying them in the process. We applied this scenario to a breakfast situation in which a cobot supports a person with physical impairments in performing basic tasks such as picking up a bottle and pouring a glass of water.

We used two established off-screen visualization techniques from research on small screens, namely *Wegde* [24] and *Halo* [25], where off-screen guidance is a well-explored topic. We added a third visualization technique *Line* (see Figure 1), which aims to reduce the potential drawback of visual clutter of *Wegde* and *Halo* as well as reduce the level of detail encoded in the visualization. All three approaches are used to continuously communicate the position of each object as they are perceived by the cobot. The user should immediately recognize a failure in object detection as the visualization for the lost object ceases.

We conducted two remote user studies exploring the efficiency, effectiveness, and task load of all three off-screen visualizations when communicating robot perception. First, we provided an exploratory experiment with 12 participants from our target user group of people with physical impairments. Second, we followed up with a validation experiment with 116 participants without physical impairments. Both studies show that a simple but clear visualization approach such as *Line* provides advantages for robot perception communication, both in terms of user preferences as well as objective measures. The remote nature of our study was adapted (a) to accommodate for social distancing guidelines during the SARS-CoV-2 pandemic; and (b) to allow for a more controlled and risk-reduced setup for target group participants.

## 2. Related Work

Previous literature has focused on (a) the usage of cobots for care support; (b) AR in human–robot collaborations; and (c) visualization techniques for target localization. We focus on ways cobots can effectively communicate their perception.

### 2.1. Cobots for Care

In 2021, the World Health Organization estimated that 15% of people live with some form of disability. (WHO. Disability & Health Report. https://www.who.int/news-room/fact-sheets/detail/disability-and-health, last retrieved 30 December 2021). Building on this, 7.9 million people are classed as severely disabled in Germany alone. (DESTATIS. Disability Facts and Figures—Brief Report 2019. https://www.destatis.de/DE/Themen/Gesellschaft-Umwelt/Gesundheit/Behinderte-Menschen/Publikationen/Downloads-Behinderte-Menschen/sozial-schwerbehinderte-kb-5227101199004.html, last retrieved 30 December 2021). Over 58% of these cases cover people with physical impairments and therefore we focused on this group for our study. In particular, we concentrated on people with a permanent and significant degree of compromised mobility of the extremities. Ample literature has examined the impact of assistive robotic systems in supporting people with motor impairments. The works of Chen et al. [5] for the *Robots for Humanity* project and Fattal et al. [6] looked into the feasibility and acceptance of robotic systems as assistive technologies. Both found that robotic devices are often designed to assist with several different activities of daily living, often resulting in larger robotic devices that frequently require a robotic arm mounted on a mobile unit. Drolshagen et al. investigated the acceptance of robots in sheltered workshops, finding that robots are quickly accepted and close proximity is preferred [26]. Currently, a trend can be observed towards the research into cobots in domestic care to support people in their everyday lives [27,28,29,30,31,32]. One elementary part of everyday tasks is the consumption of food and drinks [7]. Based on this, the present study investigates cobot assistance for people with physical impairments during a standard breakfast scenario.

### 2.2. Augmented Reality in Human–Robot Collaboration

In recent decades, AR technology has been frequently used for human–robot collaboration [33]. Previous work has mainly focused on the use of HMDs, MAR, SAR and the visualization of the robot motion intent [9,34,35]. Rosen et al. showed that AR is an improvement compared to classical desktop interfaces when visualizing the intended motion of robots [34]. However, while visualizations of motion intent have been studied extensively in previous work [9,10,11,12,13,14], communicating cobot perception remains an open challenge. It is vital that the human user can recognize what the semi-autonomous robotic system perceives explicitly (e.g., objects such as a glass or a bottle) because this enables users to recognize any occurred error in the robot’s perception [16,17]. The non-perception of objects can have especially drastic consequences, as often demonstrated in autonomous driving. (Guardian. https://www.theguardian.com/technology/2018/mar/19/uber-self-driving-car-kills-woman-arizona-tempe, last retrieved 30 December 2021) In this paper, we focused on the communication of cobot perception, and in particular, different methods to make the cobot’s sensor-based detection of objects in its surroundings visible and clear to the user.

### 2.3. Visualization Techniques for Object Localization

As discussed in Section 1, AR, and in particular HMD-AR or MAR, reduce the field of view of the user as they either only display information in a small part in front of the user’s eye (HMD) or on the available screen area. This means that they often require guiding the users’ attention to an off-screen object of interest. For example, Biocca et al. proposed the *Attention Funnel* to achieve this attention shift [36]. However, these mostly depend on the possibility of MAR and HMD approaches to easily adjust the field of view by turning the device or head. Adapting them to SAR might be difficult as the projection is often fixed. In addition, these approaches are not usually meant to highlight and identify multiple objects in the surroundings. They would likely overwhelm the user with too much visual clutter.

However, there is a large body of research in the context of off-screen visualization techniques, originally addressing the challenge of small-screen devices, which could pose a promising approach for this particular challenge. *Halo* is an early off-screen visualization method proposed by Baudisch et al. and initially intended for small, rectangular screens [25]. It uses circles with their center around off-screen objects and their radius just large enough to cut the screen’s border. Using *Halo*, the distance information is encoded in the arcs themselves and directly incorporates the scale of the scene, which was preferred by the users. Furthermore, *Halo* can be extended to on-screen objects by drawing the circle around the object. *Wedge* is another frequently used off-screen visualization technique proposed by Gustafson et al. [24]. It visualizes off-screen objects by attaching isosceles triangles to them. Two corners of the triangle are always on screen; the third is fixed to the point of interest. This leads to an encoded distance information by an amodal completion as with *Halo*. Similarly to *Halo*, *Wedge* can also be used to visualize in-view objects. Gruenefeld et al. already demonstrated in two studies that both *Halo* and *Wedge* are transferable to AR; however, they did not investigate SAR [37,38]. To address this, we applied off-screen visualization techniques to SAR and investigated their effectiveness for conveying cobot perception by visualizing all objects currently detected by the sensor system.

## 3. Experimental Approach

In this paper, we investigated how to communicate cobot perception in a scenario related to activities of daily living (ADL). Our main target group are people with physical impairments. Our previous work—an ethnographic study to establish recommendations for the development of a robotic drinking and eating aids—has shown a clear need for (semi-) autonomous assistive technology during meal time [7]. Hence, we focused on a breakfast situation at a kitchen table (see Figure 2a). Our goal is to help users understand the cobot and its actions so that users are able to understand how the cobot works and predict potential failures. Overall, this should contribute to better collaboration and foster trust and acceptance.

Our research investigates the subjective experience, effectiveness, and efficiency of different visualization approaches. We conducted two independent remote studies with 12 participants in the first and 116 participants in the second experiment, analyzing a total of three different visualizations. In the first experiment (see Section 4), 12 people with physical impairments—the target group—participated and delivered valuable quantitative and qualitative insights. After this, we conducted a second experiment (see Section 5 Experiment II: Validation Study) with 116 participants without impairments to verify our findings. We selected the established visualizations *Halo* and *Wedge* (see Section 3.2.3) and compared them to a simplified line-based visualization—*Line*.

All visualizations served the purpose of optically highlighting and indicating each object on a kitchen table that the sensory system of the cobot is currently detecting. The challenge for the user is to understand a potential failure of the system, indicated by the ceased highlighting of a previously perceived object. The studies required users to recognize these failures and indicate the object no longer detected by the cobot.

We applied a SAR solution using a projector to display visualizations on the table surface, as detailed in Section 3.2. This visualization technology enables a dynamic workspace of the cobot with visual cues directly projected in the working area of the kitchen table.

### 3.1. Experimental Task

We wanted to determine which visualization technique allows users to recognize the cobot’s perception errors quickly, accurately and with minimal effort. Therefore, users were presented with a simple task (see Section 4.4 and Section 5.4). They had to observe a virtual scene where a robot arm was moving across a breakfast table containing multiple items. Initially the current visualization technique shows each object as detected and perceived by the cobot. After a randomized time in an interval of 5–15 seconds in *experiment I* (see Section 4) and an interval of 3–15 seconds in *experiment II* (see Section 5), the cobot ceased to detect a random object; indicated by a vanished visualization. The user had to a) recognize this situation as quickly as possible and b) identify the no longer perceived object.

### 3.2. Apparatus

Here, we describe the developed apparatus of both experiments. In particular, we (a) describe our 3D testbed environment; (b) compare different mounting settings of the projector and report the concluding setting; and (c) introduce the selected visualization techniques.

#### 3.2.1. 3D Testbed Environment

We developed a simulation of the robot setup present in our laboratory using the Unity3D Game Engine. (https://unity.com/, last retrieved 30 December 2021) We used Bio IK (https://assetstore.unity.com/packages/tools/animation/bio-ik-67819, last retrieved 30 December 2021) to simulate the robot’s inverse kinematics. The project was exported as a WebGL application and hosted online for easy access by the participants within their particular web-browser environment. Any user interaction with the prototype was performed through mouse clicks, enabling participants with motor impairments to use their respective pointing devices.

For the virtual robot, we used a simulated KUKA LBR iiwa 7 R800 robot with a Robotiq 2-Finger 85 gripper module attached to the robot’s flange. A simulated projector connected to the virtual flanch of the gripper/robot points towards the object of interest. Alternative mounting positions of the projector are discussed in Section 3.2.2. A virtual plane with a circular cutout restricts the simulated projection radius, creating a circular shape of projection to ensure the same size of projection to every site (projection distance: 50 cm; projection radius: 15 cm). The robot is located in front of a table (dimensions: 120 cm × 60 cm × 75 cm) with five items one might find in a hypothetical breakfast scenario (a box of cereals, a carton of milk, a plate of fruits, a bowl and a mug). See Figure 2 for a glance at the setup and a closeup on the projection.

#### 3.2.2. Different Mounting Settings of the Projector

As part of the projection-based cobot perception visualization development process, we compared different potential mounting options for a pico-sized projector in a real-world setting with a real robot. As illustrated in Figure 3, we compared a top-mounted projection, e.g., from the ceiling, with a side-mounted projection, e.g., by using a tripod, with a cobot flanch-mounted projection by attaching it next to the gripper.

**Top-mounted projection**: Because of the large distance between the projector and table surface, a top-mounted projection has a large area. It can cover the whole workspace, e.g., the surface of the table, to visualize cobot perception. However, as shown in Figure 3a, objects in the vicinity of the cobot arm are not visualized. In addition, any visualization trying to highlight objects directly beneath the gripper is not visible due to the shadow that is cast by the cobot device itself.**Side-mounted projection**: Attaching the projector at one side of the table tackles this issue of visualizing the objects of interest beneath the gripper and also enables a quite large projection area (see Figure 3b). However, a shadow can still hide the visualization related to objects in the vicinity of the cobot’s arm due to the same reasons.**Cobot flanch-mounted projection**: By mounting the pico-projector to the cobot’s flanch next to the gripper, the cobot or its gripper do not cast a shadow within the projection area. Because the light comes from above, the size of the objects’ shadows is reduced in contrast to the other projector settings, as shown in Figure 3c. As a drawback, the projection area is limited, and therefore, increases the need for visualization which can also highlight off-screen objects, which are currently not placed within the projection area.

A top- or side-mounted setting leads to a large projection area but casts a shadow that a visualization cannot overcome. Here, it is especially hard to take into account the shadow cast by the robot arm and compensate for its’ impact on a presented visualization. Whereas a flanch-mounted setting enables visualizing the objects that need the most attention in the case of cobot failure—the objects right under the gripper—but reduces the projection area. However, as discussed, we aim to explore whether this issue can be tackled by using off-screen visualization techniques.

#### 3.2.3. Selected Visualization Techniques

In our user study, we compare three different visualization methods: (a) *Halo*; (b) *Wedge*; and (c) *Line*. *Halo* and *Wedge* are well-established off-screen visualization techniques taken from previous work (see Section 2.3), while *Line* is proposed by us (see Figure 4).

**Halo:** Off-screen objects are visualized by attaching circles around objects, which any person around the table can see as well. These circles are always drawn with a radius as large as necessary to visualize part of the circle in the on-screen area. This means that the user can (a) understand the direction of the target object in the off-screen area; and (b) determine the distance, as this is encoded through the radius of the circle. For on-screen objects, we kept the circle visualization and show the radius to be 5 cm larger than the radius of the object’s footprint.**Wedge:** While the approach works similarly to *Halo*, here, off-screen objects are visualized by attaching isosceles triangles to them. Two corners of the triangle are always on-screen; the third is fixed to the point of interest. The distance is encoded via an amodal completion of the triangle, which avoids overlapping and leads to a reduced visual clutter. This allows a more accurate determination of the object’s distance compared to *Halo*. For on-screen objects, we decided to keep the triangle attached to the corner of the object, pointing towards its center. The on-screen triangles point from the projection center to the object’s center, comparable to arrow-based techniques [37].**Line:** While *Halo* and *Wedge* try to encode distance information quite accurately, they also lead to visual clutter when many off-screen objects are visualized at the same time. Therefore, as a baseline, we propose a reduction to a simple line-based visualization technique. Here, a line connects the center of the projection to each object’s center. Several lines—for each object one—are “shining” in a manner resembling a beam from the center in every direction (see Figure 2b). We still encode some distance information through the light intensity of the *Line* visualization.

As previous research shows that both *Wedge* and *Halo* work well within the realm of small-screen devices, we wanted to explore whether this can be adapted to the presented SAR off-screen problem. Visual complexity, in general, could cause problems when rapid judgments are necessary. Given that users need to recognize errors quickly and accurately, we wanted to add a visually less complex visualization method with *Line*. Still, all visualizations allow the user (a) to see that an object is recognized and (b) to infer its position relative to the projection center.

## 4. Experiment I: Target Group

The first experiment compared different visualization techniques following our experimental approach (see Section 3). We involved participants of the target group—people with physical impairments. Our goal was to explore how to best communicate cobot perception feedback to potential users for such essential tasks such as having breakfast to enable a more independent and self-determined life.

### 4.1. Study Design

To evaluate the performance of different visualization techniques for conveying cobot perception, we conducted a within-subjects remote user study with an counterbalancing order of the visualization techniques. Our independent variable was the visualization technique with three levels (*Line* vs. *Wedge* vs. *Halo*).

As dependent variables, we used a mixed-methods approach. As quantitative measures to evaluate task performance, we took into account recognizability, accuracy, reaction time, task load and individual Likert-scaled items. Furthermore, we collected qualitative data in the form of subjective feedback from our participants.

The recognizability describes the percentage of how often the simulated cobot failure was correctly recognized. To analyze this, we counted the number of trials on which a participant clicked—to interrupt the trial—only after an actual cobot failure happened. We acknowledge that this measure only gives an indication of recognizability, as we cannot be sure whether the participants really recognized the failure or simply thought that it should already have happened. Still, it excludes those clicks that happened before a failure occurred, where we can be sure that participants did not correctly judge the situation.

The accuracy describes the percentage of how often the correct affected object was identified. To determine the accuracy, we compared the selected object by the participants that they thought was no longer perceived by the cobot with the correct one. The result could either be correct or incorrect (0, 1).

For the reaction time, we measured the time from which a cobot’s perception error happened to the point in time when the participant performed a mouse click (or equivalent input device). To reduce the impact of individual differences on reaction time, which can be quite large given not just cognitive differences but also differences in input devices and physical abilities, we measured a baseline reaction time for each participant and subtracted the median of this testing from the individual measurement. The resulting reaction time is: $time_{reaction}=time_{clicked}-time_{failure}-median(t_{reactionPreTest}).$

We used the mean of the task load scores by dimensions as measured by the NASA Raw-Task Load Index (Raw-TLX) [39] to determine the participants’ perceived task load during the trials.

After each visualization, we asked three 7-point Likert-items (1 = strongly agree, 7 = strongly disagree) to determine participants’ ability to detect which objects were perceived by the cobot and which were not, that neither the cobot itself nor the number of objects made it hard to observe the scenario, and if visualizations were understandable. We reported the mean values of each 7-point Likert-item.

We asked our participants to sign up for post-test interviews if they were interested. Unfortunately, only two of the participants did so. We conducted a 25-min telephone interview with these two participants on the same day that they participated in the remote experiment. Here, eleven open-ended questions were asked about the following topics:Status quo and acceptance of technology support;Appearance and implications;Trust and understanding;Preference and reason;Importance of a perceptual feedback;

### 4.2. Research Questions

To explore the suitability of the three selected visualizations, our research was guided by the following set of research questions:$RQ_{1}$**Do*****Wedge*****and*****Halo*****—because of their more detailed integrated distance information—enable the more accurate identification of failure objects or does the extra visual clutter disturb the user?**$RQ_{2}$**Do the different visualization techniques have an influence on the reaction time, i. e., are certain visual features quicker to recognize, process and thereby identify when they vanish?**$RQ_{3}$**How do users recognize the task load of different visualizations? Is the extra visual clutter of*****Wedge*****and*****Halo*****considered a problem or does the integrated distance information actually help reduce the task load?**

### 4.3. Participants

Twelve volunteers participated in this experiment: three females, three males and six who preferred not to say. They fell into four age groups: two participants were aged between 30 and 39 years; three participants were aged between 40 and 49 years; two participants were aged between 50 and 59 years; and one participant was aged between 60 and 69 years. Four participants preferred not to state their age. All participants suffered from complex motor impairments caused by spinal cord injuries and required assistance in everyday life. Only one participant had prior experience with cobots, while three participants mentioned some experience with toy robots.

Participants were recruited via announcements in different social media communities regarding assistive technology (e.g., Paraplegie.ch, Assistive Technology Community. https://community.paraplegie.ch/de/forum/hilfsmittel-technologie, last retrieved 30 December 2021) and social media discussion communities for people suffering from multiple sclerosis (MS) (e.g., mein.ms-life, Community for people suffering from multiple sclerosis (MS). https://mein.ms-life.de/ms-community/, last retrieved 30 December 2021) among other more local announcements. Participants did not receive any monetary compensation.

### 4.4. Procedure

Before the experiment started, participants were informed about the study and the experimental setup. This was implemented as a landing page for the study’s URL. Participants had to give their informed consent by enabling a checkbox. Through another checkbox, they gave us the permission to use their anonymized recorded data. After a short demographic questionnaire, participants performed a reaction time test. We measured their reaction time when clicking on a screen as soon as a change in display color occurred. Ten repetitions allowed us to define the median time needed for the participant to react to a stimulus. We used this datum to determine the actual reaction time after recognizing a cobot perception error, thus reducing variability between subjects because of individual differences (e.g., latency of input devices, differences in physical abilities).

Participants then viewed a screen describing the first visualization method. We used images highlighting and describing any part of the visualization and a full text which gave step-by-step instructions. In a subsequent trial run, they watched the cobot perform a set of movement paths—which differed from those in other trials. Participants were instructed to click anywhere on the screen as soon as they noticed the disappearance of a visualization connected to an object. Right after they did click on such a case, a screen appeared which showed all potential target objects next to each other. The participants could then choose the item they thought the cobot did no longer perceive without any time constraints. Once this trial run was completed, the cobot performed twelve different movement paths as repetitions of this task, counting towards the data analysis. Participants viewed the twelve pre-programmed paths in random order. Six paths had an on-screen object disappear and six paths had an off-screen object disappear. Objects disappeared after a random time of between 5 and 15 s.

Once they completed all twelve paths for one visualization, participants filled out a NASA Raw-TLX questionnaire to report their workload. They also answered three additional questions specifically tailored to the respective experiment to evaluate their preferred visualization. The entire process was repeated with the two remaining visualization methods. The order in which the three visualization types were shown was counterbalanced using a Latin-square design. This experiment lasted an average of 40 min. The two participants who volunteered for the post-test interview did take part in this, as stated, on the same day as participating in the online study.

### 4.5. Results

During the *Line* technique run, one participant did not generate valid reaction times, as they performed mouse clicks before the cobot failure actually happened in every single trial. While this may be caused by an ineffective visualization, the fact that this happened in each trial and usually instantaneously after the start led us to the conclusion that the participant did not follow the test protocol. Consequently, this participant was excluded, which resulted in 11 remaining valid participant responses. We did not assume normality for the statistical analysis of our quantitative data and therefore we relied on non-parametric tests. Given the within-subject design with three conditions, we first applied a Friedman test as omnibus test followed by Wilcoxon tests as post hoc pairwise analysis with Bonferroni–Holm correction applied. Overall, the experiment resulted in 396 (11 participants × 3 visualization techniques × 12 trials) measured trials excluding training trials. Used abbreviations and symbols are:SD: Standard deviation;$\chi^{2}$(2): Chi-squared with two degrees of freedom;*p*: *p*-value as expression of the level of statistical significance (*p*: ≤0.05 *, ≤0.01 **, and ≤0.001 ***);N: Sample size;W: Minimum sum of ranks;Z: Normalized minimum sum of ranks;r: Effect size (r: >0.1 small, >0.3 medium, and >0.5 large effect).

#### 4.5.1. Recognizability: Percentage of Correctly Recognized Cobot Failures

In each trial, the cobot failed after 5–15 seconds. Participants had to respond with a mouse click to verify that they recognized the failure. However, in certain trials, participants did not click at all (*Wedge* = 3 trials; *Halo* = 2 trials; *Line* = 0 trials) or clicked before the object disappeared (*Halo* = 11 trials; *Wedge* = 3 trials; *Line* = 1 trials). From the reaction test at the beginning of the experiment, we calculated a median of the reaction time for each participant. This was taken into account to count those trials as unsuccessful, when the individual reaction to a cobot failure was faster than the median reaction time (*Line* = 12 trials; *Halo* = 11 trials; *Wedge* = 8 trials). The mean percentage of correctly recognized trials per participant for each visualization are (in descending order): *Line* = 90.2% (SD = 19.7%); *Wedge* = 89.4% (SD = 10.6%); and *Halo* = 81.8% (SD = 17.8%). A Friedman test showed no significant main effect of percentage of recognized failures on visualization ($\chi^{2}$(2) = 4.71, *p* = 0.095, N = 11).

Moreover, we can distinguish between correctly recognized on-screen and off-screen objects. The mean percentage of correctly recognized on-screen objects for each visualization are (in descending order): *Line* = 92.4% (SD = 20.2%); *Wedge* = 87.9% (SD = 15.1%); and *Halo* = 78.9% (SD = 27.0%). A Friedman test showed no significant differences ($\chi^{2}$(2) = 4.26, *p* = 0.119, N = 11). The mean percentage of correctly recognized off-screen objects for each visualization are (in descending order): *Wedge* = 90.9% (SD = 11.5%); *Line* = 87.9% (SD = 19.8%); and *Halo* = 84.8% (SD = 13.9%). A Friedman test again showed no significant differences ($\chi^{2}$(2) = 1.19, *p* = 0.552, N = 11).

#### 4.5.2. Accuracy: Percentage of Correctly Identified Failure Objects

For the percentage of correctly identified objects that the cobot failed to perceive during the trial, we only considered all trials for which participants responded after the cobot failure happened (n = 345) and therefore had a chance to select the correct object. The mean percentage per participant of correctly identified failure objects per visualization are (in descending order): *Line* = 94.4% (SD = 7.1%); *Wedge* = 77.8% (SD = 15.9%); and *Halo* = 72.5% (SD = 19.9%). A Friedman test showed a significant main effect ($\chi^{2}$(2) = 8.72, *p* = 0.012 *, N = 11). Post hoc pairwise comparisons using a Wilcoxon signed-rank with Bonferroni correction showed a significant difference between *Halo* and *Line*, but not between any other pairs (see Table 1).

#### 4.5.3. Reaction Time

For the reaction time, we only considered trials for which participants correctly responded, meaning that participants clicked after the failure happened, responded before the trial ended (10 seconds after the cobot failure happened) and clicked on the correct object (n = 282; *Line* = 112, *Wedge* = 92, *Halo* = 78). We measured the time from the failure of the cobot visualization to the participant’s mouse click. Again, we considered the median reaction time from the reaction test during the beginning of the study. The mean reaction time per visualization without extreme outliers (≥3 × IQR) was calculated for each participant.The mean reaction times for each visualization calculated over the means of the participants (without values ≥3 × IQR) are (in ascending order): *Line* = 1.11 s (SD = 1.05 s); *Wedge* = 2.51 s (SD = 1.47 s); and *Halo* = 4.26 s (SD = 2.63 s). The reaction times are plotted in Figure 5.

A Friedman test revealed a significant main effect of reaction time on visualization ($\chi^{2}$(2) = 11.64, *p* = 0.003 **, N = 11). Post hoc pairwise comparisons using a Wilcoxon signed-rank with Bonferroni correction showed a significant difference between *Halo* and *Line*, but not between any other pairs (see Table 2). Concerning reaction times, we can conclude that *Line* has a significant lower reaction time than *Halo*.

#### 4.5.4. Task Load

The mean of the task load ratings as measured by the NASA Raw-Task Load Index (Raw-TLX) [39] are (in ascending order): *Line* = 22.89 (SD = 16.47); *Halo* = 40.83 (SD = 17.28); and *Wedge* = 47.13 (SD = 22.89). A Friedman test revealed a significant main effect of task load on visualization ($\chi^{2}$(2) = 7.09, *p* = 0.029 *, N = 11). Post hoc pairwise comparisons using a Wilcoxon signed-rank with Bonferroni correction showed a significant difference between *Wedge* and *Line* (W = 60, Z = 2.40, *p* = 0.041 *, r = 0.51) and *Halo* and *Line* (W = 56, Z = 2.04, *p* = 0.042 *, r = 0.44), but not between *Wedge* and *Halo* (W = 44, Z = 0.98, *p* = 0.730, r = 0.21). Concerning the task load, we can conclude that *Line* has a significantly lower task load than *Wedge* and *Halo*. The resulting task load scores per individual dimension of the TLX are presented in Figure 6.

**Mental demand:** A Friedman test revealed no significant main effect of mental demand on visualization ($\chi^{2}$(2) = 15.69, *p* = 0.058, N = 11);**Physical demand:** A Friedman test revealed no significant main effect of physical demand on visualization ($\chi^{2}$(2) = 5.43, *p* = 0.066, N = 11);**Temporal demand:** A Friedman test revealed no significant main effect of physical demand on visualization ($\chi^{2}$(2) = 4.89, *p* = 0.087, N = 11);**Performance:** A Friedman test revealed a significant main effect of physical demand on visualization ($\chi^{2}$(2) = 8.79, *p* = 0.012 *, N = 11). Post hoc pairwise comparisons using a Wilcoxon signed-rank with Bonferroni correction showed a significant difference between *Halo* and *Line* (W = 66, Z = 2.93, *p* = 0.003 **, r = 0.63), but not between *Wedge* and *Halo* (W = 21.5, Z = −1.03, *p* = 0.668, r = 0.22) and *Halo* and *Line* (W = 39, Z = 1.20, *p* = 0.744, r = 0.26);**Effort:** A Friedman test revealed no significant main effect of physical demand on visualization ($\chi^{2}$(2) = 3.21, *p* = 0.201, N = 11);**Frustration:** A Friedman test revealed a significant main effect of physical demand on visualization ($\chi^{2}$(2) = 7.39, *p* = 0.025 *, N = 11). Post hoc pairwise comparisons using a Wilcoxon signed-rank with Bonferroni correction showed a significant difference between *Halo* and *Line* (W = 45, Z = 2.82, *p* = 0.012 *, r = 0.60), but not between *Wedge* and *Halo* (W = 17, Z = −0.76, *p* = 0.977, r = 0.16) and *Halo* and *Line* (W = 26, Z = 0.99, *p* = 0.703, r = 0.21).

#### 4.5.5. Individual Likert-Items

After each visualization, we asked 3 7-point Likert-items (1 = strongly agree–7 = strongly disagree). Participants stated that the visualization helped them understand the position of the objects on the table for *Line* (Md = 2, IQR = 2.5) while they slightly disagreed for *Wedge* (Md = 5, IQR = 1.5) and *Halo* (Md = 5, IQR = 1.5). A Friedman test showed a significant main effect ($\chi^{2}$(2) = 6.45, *p* = 0.040 *, N = 11). Post hoc pairwise comparisons using a Wilcoxon signed-rank with Bonferroni correction showed a significant difference between *Wedge* and *Line* (W = 53.5, Z = 2.69, *p* = 0.018 *, r = 0.57, but not between *Wedge* and *Halo* (W = 20, Z = 0.18, *p* = 0.891, r = 0.04) and *Halo* and *Line* (W = 55.5, Z = 2.02, *p* = 0.146, r = 0.43).

Moreover, participants voiced that they could easily notice the cobot failure for *Line* (Md = 2, IQR = 2), while they slightly disagreed *Halo* (Md = 5, IQR = 2) and *Wedge* (Md = 6, IQR = 2). A Friedman test revealed a significant main effect ($\chi^{2}$(2) = 9.5, *p* = 0.009 **, N = 11). Post hoc pairwise comparisons using a Wilcoxon signed-rank with Bonferroni correction showed a significant difference between *Wedge* and *Line* (W = 45, Z = 2.83, *p* = 0.012, r = 0.60), but not between *Wedge* and *Halo* (W = 11.5, Z = 0.09, *p* = 0.969, r = 0.02) and *Halo* and *Line* (W = 49, Z = 2.19, *p* = 0.094, r = 0.47).

Furthermore, participants stated that for them, the cobot itself does not interfere with the recognition of the visualizations for *Line* (Md = 3, IQR = 2). In contrast, they slightly disagreed for *Wedge* (Md = 4, IQR = 1) and *Halo* (Md = 5, IQR = 1.5). A Friedman test revealed a significant main effect ($\chi^{2}$(2) = 7.09, *p* = 0.029 *, N = 11). Post hoc pairwise comparisons using a Wilcoxon signed-rank with Bonferroni correction showed a significant difference between *Halo* and *Line* (W = 43.5, Z = 2.55, *p* = 0.035 *, r = 0.54), but not between *Wedge* and *Halo* (W = 2.5, Z = −1.98, *p* = 0.234, r = 0.42) and *Wedge* and *Line* (W = 29.5, Z = 0.90, *p* = 0.867, r = 0.19).

#### 4.5.6. Qualitative Insights

We applied open coding, followed by a thematic analysis of our interview data. We did this to find patterns of two participants’ opinions and thoughts about the cobot assistance and presented visualizations. Once all the interviews were completed, two researchers transcribed all audio recordings and open coded the transcriptions. We then conducted an online affinity diagram of the open codes and organized the codes into groups, using Miro (https://miro.com, last retrieved 30 December 2021)—an online whiteboard [40]. During the telephone interview, we asked 11 open questions covering the status quo, a need for assistive technology, trust against such a cobot, if visualizations can increase this trust and understandability of cobot’s perception, and which visualization technique they would or would not prefer and why. Because we asked open questions, we could identify further insights from the participants in addition to the one related to the visualizations. We identified four main themes, which we outline below.

##### Scenario and Technology Support

Both participants relied on assistance during breakfast from their caregivers and were interested in the concept of a cobot-supported breakfast routine. P2: “I am entirely open-minded and always interested in trying new things”. However, several concerns were voiced, including the worry about the cost of a robotic aid and replacing the human caregiver, thus resulting in decreased social interaction. One additional design feature was frequently requested: the ability to mount the robotic arm to a wheelchair to increase flexibility.

##### Trust and Understanding

From the onset, the overall trust towards a robotic aid was high. However, the same principles as with humans apply; trust has to be earned. Participants indicated that their confidence in a cobot increased when they observed the cobot’s perception and communicated with it. P1: “I understood the cobot’s perception visualization, which helped me trust the system because I could see which objects were perceived by the cobot”. Easy-to-understand visualization methods can help address this concern by clearly displaying the cobot’s perception, allowing for a greater level of user oversight.

##### Positive Feedback

Post-experiment feedback was positive. Participants were generally happy with the overall look and appearance of the cobot, a frequent concern of potential users. The off-screen visualization helped increase trust and user acceptance by increasing the collaborative effort between the human and cobot. Participants preferred the proposed new *Line* type over traditional visualization methods. P1: “I liked *Line* the most because there was always a clear reference, and I could see when something was out of order, even when I was looking somewhere else”.

##### Problems and Drawbacks

When designing for non-tech-savvy users and physically vulnerable people, great care must be taken that the technology works consistently before releasing it for general use. P1: “I assume that any teething problems have been removed beforehand. That is why I already have a certain basic trust”. This also increases end-user acceptance and addresses frequently mentioned reservations concerning the cobot making more mistakes than a human caregiver. P2: “I only have my caregivers as a reference. So if the cobot does not knock something over more often than my caregivers, I would be happy. However, even if the cobot would make a few more mistakes, I could forgive the cobot”.

Participants voiced several issues concerning possible communication methods. Both *Wedge* and *Halo* were regarded as excessively complex and difficult to understand. P1: “I had problems recognizing *Wedge* because it was difficult to interpret the truncated arrows correctly”. P2: “I did not prefer the visualization with the circles [*Halo*]. I really could not distinguish anything, and when something was gone, I could only guess which object it was”.

### 4.6. Discussion

The results show a clear overall preference for a simple visualization technique such as *Line* to communicate which objects are recognized by the cobot. More complexly shaped visualizations, such as *Wedge* and *Halo*, lead to a comparatively higher task load in detecting unperceived objects.

#### 4.6.1. Performance of Visualizations

No statistically significant results were found concerning correctly perceived cobot failures with overall high detection rates. This indicates that all three visualizations were effective in communicating the failure states.

However, there were differences in efficiency, with *Line* showing a significantly lower reaction time than *Halo* but not *Wedge*. Interestingly, the same results apply for accuracy, as *Line* shows a significantly higher accuracy compared to *Halo*. This indicates that the simple coding of *Line* has benefits even when the user has to understand which object is affected. The added information of the distance coding of Halo does not seem to overcome potential limitations due to visual clutter. While descriptive data do show differences between *Line* and *Wedge*, statistical tests do not confirm this, potentially due to low statistical power in a study with only twelve participants.

In our study, we were unable to confirm the advantages of *Wedge* in contrast to *Halo* in error rate and completion time as mentioned by Gustafson et al. [24]. While the descriptive data do show differences, another reason might be the round projection area. One potential advantage of *Wedge* compared to *Halo* is to overcome the corner-density problem of the latter [25], which is not applicable to round shaped-screens. This is in line with findings from Gruenefeld et al., who also found that with a round visualization area, the advantage of *Wedge* over *Halo* is less strong [37].

#### 4.6.2. Task Load

We found that *Line* could significantly reduce participants’ task load compared to the more complex *Wedge* and *Halo*. We believe this to be due to the simpler shape of *Line*, resulting in less visual clutter. In contrast to the other visualizations, with *Line*, users’ focus of attention is on the gripper and center of the projection, which allows them to directly observe any changes. Using *Line* does not require the user to observe the periphery, making the visualization easier to interpret. Special consideration must be given to design an easy-to-view visualization, with objects and paths large enough to be identified at a glance.

#### 4.6.3. Usefulness and Trust

Qualitative insights showed a clear preference for *Line*, mainly due to its avoidance of overlapping visualizations as with *Halo* and the amodal completion of the triangle as in *Wedge*. Although initial confidence in the robot’s abilities is high, this needs to be maintained through constant and consistent correct behavior and clear communication. Our participants noted that higher levels of trust develop whereas failures happen rarely. Participants highlighted that a higher rate of mistakes compared to a human caregiver would result in lower acceptance and trust in the cobot.

#### 4.6.4. Limitations

One of the main issues we faced when conducting this experiment was the small sample size. This limitation highlights the difficulty of designing for and involving people suffering from severe disabilities. The ongoing SARS-CoV-2 pandemic amplifies the problem as access to people is further restricted. Nonetheless, we believe that the remote nature of our study enabled us to gain valuable insights while granting access to participants from a wider geographic range. In addition to the low number of participants, even fewer participants were willing to participate in interviews via telephone, resulting in a limited number of qualitative data.

In our experiment, we did not measure trust with standardized questionnaires. Nonetheless, during our interviews, participants reported insight into how they would trust the cobot in this scenario. Hence, we did not address trust in-depth, i. e., with standardized measures, and thereby, our results can only be the foundation for further hypotheses and research concerning trust in cobots.

While this experiment was able to explore the potential benefits and drawbacks of the three selected visualization techniques, it did not provide clear statistical evidence in several cases. In particular, the differences between *Line* and *Wedge* did not show statistically significant effects, although *Wedge* is conceptually quite similar to *Halo*. Two difficulties can account for this, one being the overall small sample size and the second being the higher level of individual differences in the target group, potentially overshadowing smaller effects.

## 5. Experiment II: Validation Study

Based on the limitations of *experiment I* (see Section 4), we conducted a second experiment open to a non-specific user group, aiming to gain statistical evidence on particular hypotheses gained from this first experiment. Therefore, we also opted to exclude *Halo* from this second experiment, as results regarding this technique were already quite clear in the first. While the absolute results of such a second experiment with a non-specific user group may not be applicable to the target user group of people with physical impairments, we are confident that the relative results are. The main reasoning here is that the experimental task only requires very little physical interaction and the kind of physical interaction (mouse click) is kept constant for both tested visualizations *Line* and *Wedge*.

### 5.1. Study Design

Based on the study design of *experiment I* (see Section 4.1), we designed the second experiment as a within-subjects remote user study. Here, we changed, based on the results of *experiment I* (see Section 4.5), our independent variable to visualization technique with two levels (*Line* vs. *Wedge*). The order of the visualization techniques was counterbalanced. We used the same quantitative measures to evaluate task performance (recognizability, accuracy, reaction time, task load, and individual Likert-scaled items).

### 5.2. Hypotheses

Based on the results of *experiment I*, which showed the potential advantages of *Line*, we developed the following set of hypotheses:

**Hypothesis** **1**(**H1**). *We hypothesize that Line can select the failure object with higher accuracy than Wedge. Results in* experiment I *already point in this direction. It seems that the user is mostly focused on the gripper and thereby the center of the projection, which benefits Line, as any change in the visualization can be directly seen in the center. While this should first benefit the recognizability of the failure, it also seems to have a positive effect on the accuracy, as the target object can usually be inducted from the vanishing line. Thereby, this should outweigh the better distance encoding of Wedge.*

**Hypothesis** **2**(**H2**). *We expect that Line allows quicker reaction times when the cobot’s sensors lose track of an object compared to Wedge. We believe this to be the case because with Line, as before, the focus of attention is on the gripper and center of the projection, which allows the user to directly observe any changes in the Line visualization. In contrast, Wedge requires the user to observe the periphery.*

**Hypothesis** **3**(**H3**). *As a consequence of prior hypotheses, we also hypothesize that Line will lead to a lower task load. The Line visualizations that are characteristically displayed beneath the gripper and thereby in the center of the projection require less attention shifts from the user—which should be visible in task load measures.*

### 5.3. Participants

In total, we collected data from 209 participants. Since the experiment was conducted as an online study, the data of those participants were checked with regard to plausibility. We wanted to make sure that we did not include data from participants who simply ”clicked through” the study without actually following the task protocol. Therefore, as a reasonable limit, we decided to remove participants whose median time of the mouse click ($time_{clicked}$) was less than three seconds. The limit of three seconds was chosen as the task scenario was designed in such a way, that it took at least three seconds for a cobot failure to happen.

This check led to the exclusion of 93 participants. The remaining 116 participants were categorized into four age groups: 85 of them were between 18 and 29 years old; 21 of them were aged between 30 and 39 years; one was between 40 and 49 years old; and four of them were 50–59 years old; and another three were 50–59 years old. Two participants preferred not to state their age.

In total, 63 participants had used a robot before the experiment, while 53 participants had no prior experience using robots. The remaining ten participants did not mention their prior experience using robots. Among all participants, 49 had previous experience using robots in the form of toy robots. In addition, 15 participants had used drones, eight had used service robots and another seven had used industrial or humanoid robots before. Furthermore, six participants had experiences with robots other than those mentioned.

Participants were recruited via SurveyCircle (https://www.surveycircle.com/, last retrieved 30 December 2021)—an open platform for survey submissions among other more local announcements. Participants did not receive any monetary compensation, but earned “survey ranking points” for their own study on SurveyCircle.

### 5.4. Procedure

The experiment followed the same procedure as described in Section 4.4, with the only difference being that, in all twelve pre-programmed paths, only off-screen objects disappeared after a random time between 3 and 15 seconds. We changed the focus onto off-screen targets, as here lie the main conceptual differences between the visualizations.

### 5.5. Results

For the analysis, we did not assume the normality of our quantitative data, especially since the measure reaction time was not normally distributed but generally right-skewed. Other measures, such as accuracy, are dichotomous by nature and therefore not on a metric scale. As a result, we applied non-parametric statistical tests. Given the within-subject design of our evaluation, we applied Wilcoxon signed-rank tests. Overall, we had 2784 (116 participants × 2 visualization techniques × 12 trials) measured trials, excluding training trials.

#### 5.5.1. Recognizability: Percentage of Correctly Recognized Cobot Failures

The cobot failed after 3–15 seconds in each trial. Participants responded with a mouse click to verify that they recognized the failure. However, in certain trials, participants did not click at all (n = 189; *Line* = 105 trials and *Wedge* = 84 trials) or clicked before the visualization disappeared (n = 1111; *Wedge* = 601 trials and *Line* = 510 trials). From the reaction test at the beginning of the experiment, we calculated a median for the reaction time of each participant. This was taken into account to count those trials as unsuccessful, when the individual reaction to a cobot failure was faster than the median reaction time (n = 22; *Wedge* = 15 and *Line* = 7). The mean percentage of correctly recognized trials for each visualization are (in descending order): *Line* = 64.8% (SD = 29.8%) and *Wedge* = 59.5% (SD = 31.0%). A Wilcoxon signed-rank test showed no significant difference between the *Wedge* and *Line* (W = 1559, Z = −1.56, *p* = 0.120, r = 0.10, N = 116).

#### 5.5.2. Accuracy: Percentage of Correctly Identified Failure Objects

For the percentage of correctly identified objects that the cobot failed to perceive during the trial, we again only considered all trials for which participants responded after the cobot failure happened (n = 1462). The mean percentage per participant of correctly identified failure objects per visualization are (in descending order): *Line* = 72.7% (SD = 27.9%) and *Wedge* = 64.4% (SD = 30.1%). A Wilcoxon signed-rank test showed a significant difference between *Wedge* and *Line* (W = 1561.5, Z = −2.34, *p* = 0.019 *, r = 0.15, N = 116).

#### 5.5.3. Reaction Time

For the reaction time, we only considered all trials in which participants correctly responded, meaning participants clicked after the failure happened, responded before the trial ended (10 seconds after the cobot failure happened) and clicked on the correct object (n = 1100; *Line* = 608, *Wedge* = 492). We measured the time from the failure of the cobot visualization to the participant’s mouse click. Again, we considered the median reaction time from the reaction test during the beginning of the experiment. The mean reaction time per visualization without extreme outliers (≥3 × IQR) was calculated for each participant. The mean reaction times for each visualization calculated over the means of the participants (without values ≥3 × IQR) are (in ascending order): *Line* = 1.72 s (SD = 1.51 s) and *Wedge* = 2.07 s (SD = 1.81 s). The reaction times are plotted in Figure 7.

A Wilcoxon signed-rank test showed no significant difference between *Wedge* and *Line* (W = 2254, Z = 1.44, *p* = 0.151, r = 0.11).

However, it appears that there is a potential interaction effect between the order variable (start visualization) and the independent variable (visualization). When participants started with *Line*, the means are (in ascending order): *Wedge* = 2.11 s (SD = 1.94 s) and *Line* = 3.32 s (SD = 3.11 s). A Wilcoxon signed-rank test showed a significant difference between *Wedge* and *Line* (W = 413, Z = −2.17, *p* = 0.030 *, r = 0.22).

Looking at *Wedge* as the start visualizations, the means are (in ascending order): *Line* = 1.24 s (SD = 1.02 s) and *Wedge* = 2.01 s (SD = 1.64 s). A Wilcoxon signed-rank test showed a significant difference between *Wedge* and *Line* (W = 904, Z = 2.57, *p* = 0.009 **, r = 0.25).

This effect shows that the mean of *Wedge* is relatively stable, independently of it being the first or second condition participants encountered (first condition: M = 2.11 s; SD = 1.94 s and second condition: M = 2.01 s; SD = 1.64 s). A Wilcoxon signed-rank test showed no significant difference (W = 725; Z = 0.84; *p* = 0.404; r = 0.08). However, the mean of *Line* depends on the ordering of the condition (first condition: M = 3.32 s; SD = 3.11 s and second condition: M = 1.24 s; SD = 1.02 s). A Wilcoxon signed-rank test showed this difference to be statistically significant (W = 1062, Z = 4.10, *p* ≤ 0.001 ***, r = 0.41).

#### 5.5.4. Task Load

The mean of the task load ratings as measured by the NASA Raw-Task Load Index (Raw-TLX) [39] are (in ascending order): *Line* = 42.39 (SD = 15.02) and *Wedge* = 47.19 (SD = 16.03). A Wilcoxon signed-rank test showed a significant difference between *Wedge* and *Line* (W = 4483.5, Z = 3.19, *p* = 0.001 ***, r = 0.21). Concerning the task load, we can conclude that *Line* has a significantly lower task load than *Wedge*. The resulting task load scores per dimension are presented in Figure 8.

**Mental demand:** A Wilcoxon signed-rank test showed a significant difference between *Wedge* and *Line* (W = 3068, Z = 2.52, *p* = 0.011 *, r = 0.17);**Physical demand:** A Wilcoxon signed-rank test showed no significant difference between *Wedge* and *Line* (W = 1498.5, Z = 1.25, *p* = 0.211, r = 0.08);**Temporal demand:** A Wilcoxon signed-rank test showed a significant difference between *Wedge* and *Line* (W = 2900, Z = 2.24, *p* = 0.025 *, r = 0.15);**Performance:** A Wilcoxon signed-rank test showed a significant difference between *Wedge* and *Line* (W = 3618, Z = 2.35, *p* = 0.018 *, r = 0.15);**Effort:** A Wilcoxon signed-rank test showed no significant difference between *Wedge* and *Line* (W = 3158.5, Z = 1.92, *p* = 0.055, r = 0.13);**Frustration:** A Wilcoxon signed-rank test showed a significant difference between *Wedge* and *Line* (W = 3783.5, Z = 2.67, *p* = 0.007 **, r = 0.18).

#### 5.5.5. Individual Likert-Items

After each visualization, we asked 3 7-point Likert-items (1 = strongly agree–7 = strongly disagree). Participants voiced that the visualization helped them to understand whether an object on the table was not detected by the cobot for *Line* (Md = 4, IQR = 2), while they slightly disagreed for *Wedge* (Md = 5, IQR = 3). Post hoc pairwise comparisons using a Wilcoxon signed-rank showed a significant difference (W = 2857, Z = 2.28, *p* = 0.022 *, r = 0.15).

Moreover, participants stated that the number of objects on the table did not disturb them for *Line* (Md = 3, IQR = 3) and *Wedge* (Md = 3, IQR = 3). Post hoc pairwise comparisons using a Wilcoxon signed-rank showed a significant difference (W = 1808, Z = 2.31, *p* = 0.021 *, r = 0.07).

Then, in the last question, participants mentioned that the visualization was always understandable for *Line* (Md = 3, IQR = 3), while they slightly disagreed *Wedge* (Md = 4, IQR = 3). Post hoc pairwise comparisons using a Wilcoxon signed-rank showed no significant difference (W = 2654, Z = 3.31, *p* ≤ 0.001 ***, r = 0.22).

## 6. Discussion

Results from *experiment II* are in line with those from *experiment I* (see Section 4.6). Both highlight the advantages of using a straightforward visualization such as *Line* to show off-screen objects recognized by the cobot. Complex visualizations, such as *Wedge*, appear to lead to a higher amount of errors in detecting perception failures, identifying the corresponding object, and a perceived higher task load.

### 6.1. Performance of Visualizations

Both experiments found no statistically significant results concerning correctly recognized cobot failures. All visualizations appear effective in communicating the cobots’ failures. However, the recognition percentage dropped in the second experiment, as even with *Line*, participants were only able to correctly recognize cobot failures in 64.8% of the trials, compared to 90.2% in the first experiment. Potentially, the small but intrinsically motivated participant group in the first experiment did try to follow the protocol more closely. An alternative explanation might be an excessively conservative exclusion criteria (median response time smaller than 3 seconds) in *experiment II*.

When cobot failures were correctly recognized, a significant difference between *Line* and *Wedge* regarding accuracy became apparent. The percentage of correctly identified failure objects was highest with the *Line* (72.7%) visualization. This mirrors the results from *experiment I*, where the difference between *Halo* and *Line* was significant in favor of *Line*, with descriptive data showing an advantage also compared to *Wedge*. Hence, we can accept Hypothesis 1.

Regarding the reaction time, the overall trend resembles the first experiment, with *Line* having the lowest mean reaction time. However, the difference was again not statistically significant. Thus, we cannot accept Hypothesis 2. Interestingly, due to the simpler experimental design with only two conditions, we observed an interaction between the order of the conditions and the two different visualization techniques.

When participants worked with *Line* as the second visualization, the reaction time significantly improved compared to those cases in which participants started with *Line*—which, in turn, was not the case for *Wedge*). We concluded that *Line* might need a longer learning phase for participants to fully benefit from it. Why this is the case, however, remains an open question for future research.

### 6.2. Task Load

We found that *Line* significantly reduces participants’ task load compared to the visually more complex *Wedge* technique. Therefore, we can accept our Hypothesis 3. We attribute this to *Line* not requiring attention shifts but rather allowing the user to focus on the gripper at all times. It also does not require amodal completion to decode the distance information, which, as the accuracy results show, is not necessary to understand and identify which object is affected.

### 6.3. Individual Likert-Items

The results of the individual Likert-items show that *Line* received better scores than *Wedge*. The simpler design of *Line* ensures that cobot failure and the corresponding object can be better detected, making this visualization more obvious than *Wedge*. For both visualizations, the number of objects in the experiment, five, did not disturb participants. This might, of course, change with a larger amount of objects, which could be necessary for more complex scenarios.

### 6.4. Limitations

The high number of excluded participants (93) infer that a remote study setup has less oversight, potentially enticing participants not to follow the study protocol. This issue required the need for careful data cleaning. One reason could be that further guiding of the participants was not possible.

Since participants could not be observed during the experiment, we cannot say whether they ran the experiment on a suitable device and whether their full attention was on the trials. Based on the reduced level of control, we expect additional noise in our data which may overshadow certain effects. This is a common problem with remote studies, as one cannot ensure, for example, identical technical conditions for each participant. Nonetheless, we do not believe that there has been a systematic bias in our data due to this reduced level of control.

Based on the experiences from *experiment I*, we deliberately did not include more open-ended questions or even an interview apart from the 7-point Likert-scale items. Experience shows that with a remote online study, the longer it takes, the higher the chance of participants dropping out. Therefore, we decided to keep the study as short and concise as possible and focused on quantitative data.

In addition to projection orientation, special considerations regarding ambient light and background effects have to be taken into account when using SAR. Our investigation neglected these external factors as they are present independently of the projection orientation. Nevertheless, more research is required to better understand their influence.

## 7. Conclusions

We investigated the performance differences of three visualization techniques in communicating cobot perception for a Spatial Augmented Reality setup, specifically focusing on people with physical impairments as potential end-users. We were particularly interested in comparing well-established off-screen visualization techniques to a reduced and straightforward line-based visualization for perceived objects inside and outside the projection area. The first experiment focused on 11 target group participants, while 116 non-specific respondents participated in the second experiment. Both experiments analyzed and compared the effectiveness, efficiency, subjective satisfaction and task load of the visualization techniques *Halo*, *Wedge* and *Line*. While the reaction times showed only minimal differences between *Line* and the established off-screen visualization techniques, *Line* did significantly improve the percentage of correctly identified failure objects and persistently lowered participants’ task load. This result is mirrored by qualitative feedback from two target group participants, each highlighting the importance of an easy-to-understand visualization of the cobots’ perception. Overall, our results stress that communicating the cobots’ perception, including identification failures, is invaluable for assessing the overall situation and improving end-user trust. Our results generalize to similar pick-and-place workbench situations but may have limited applicability for more complex scenarios without a clearly defined environment. Overall, our findings add to a growing body of user-centered HRI literature with the overarching goal of increased user acceptance and confidence in cobots.

## Figures and Tables

**Figure 1 sensors-22-00755-f001:**
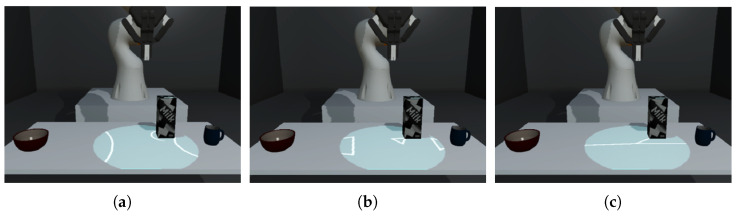
The compared visualization techniques to communicate the cobot’s perception are (**a**) *Halo*; (**b**) *Wedge*; and (**c**) *Line*.

**Figure 2 sensors-22-00755-f002:**
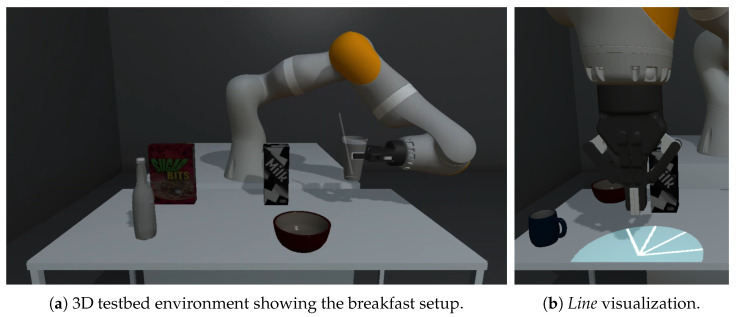
Screenshots of the 3D testbed environment. (**a**): showing the complete setup with the five items placed on the table; (**b**): showing the *Line* visualization with one object on the table not perceived by the cobot.

**Figure 3 sensors-22-00755-f003:**
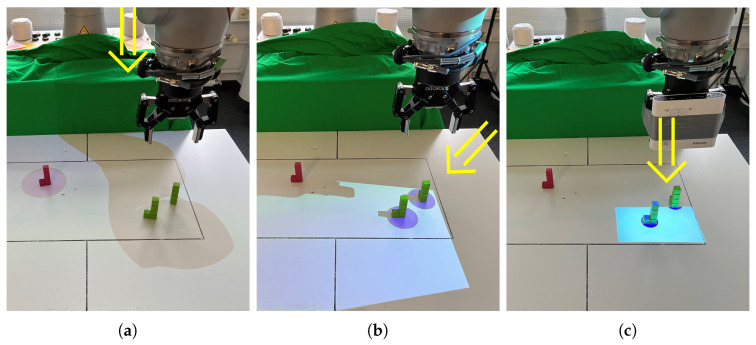
For our work, we compared three different mounting-settings of a projector in the cobot’s workspace to communicate cobot perception. The compared settings are (**a**) top-mounted; (**b**) side-mounted; and (**c**) cobot flanch-mounted projection. The direction of the projection is indicated by an arrow.

**Figure 4 sensors-22-00755-f004:**
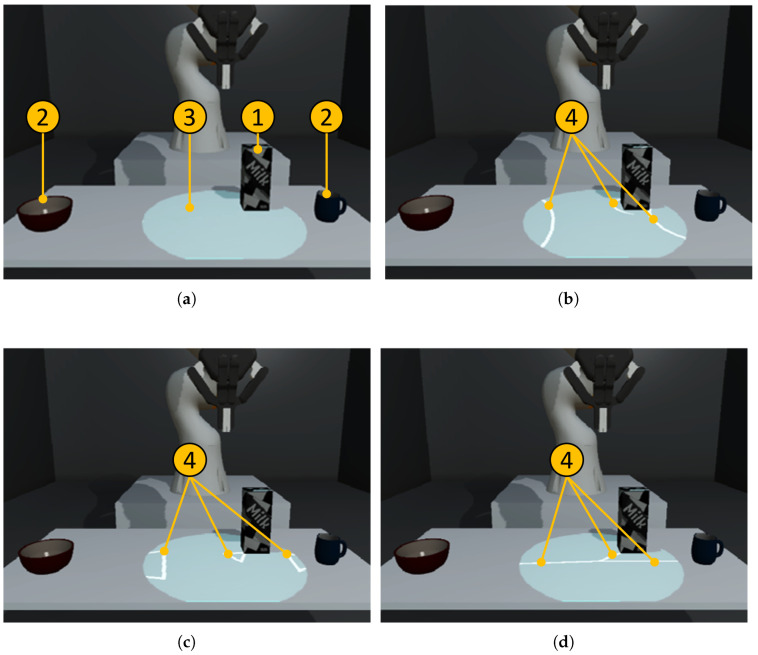
A detailed overview of (**a**) the setup highlights the different parts as (1) on-screen object; (2) off-screen object; and (3) the projection area. The (4) main features are highlighted for the selected visualization techniques (**b**) *Halo*, (**c**) *Wedge*, and (**d**) *Line*.

**Figure 5 sensors-22-00755-f005:**
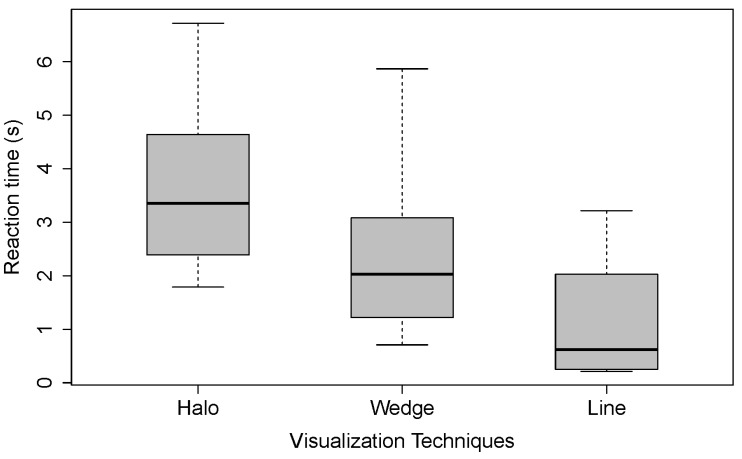
Comparison of the reaction times for the three different visualization techniques: *Wedge*; *Halo*; and *Line*.

**Figure 6 sensors-22-00755-f006:**
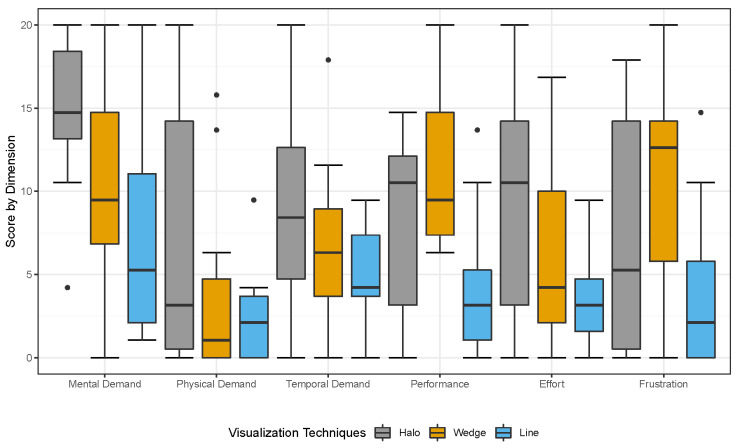
Comparison of the task load dimensions for the three different visualization techniques: *Wedge*; *Halo*; and *Line*.

**Figure 7 sensors-22-00755-f007:**
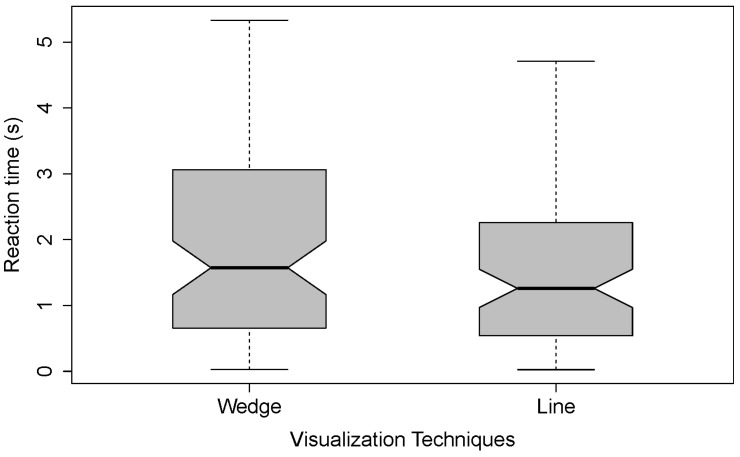
Comparison of reaction times for the two different visualization techniques: *Wedge* and *Line*.

**Figure 8 sensors-22-00755-f008:**
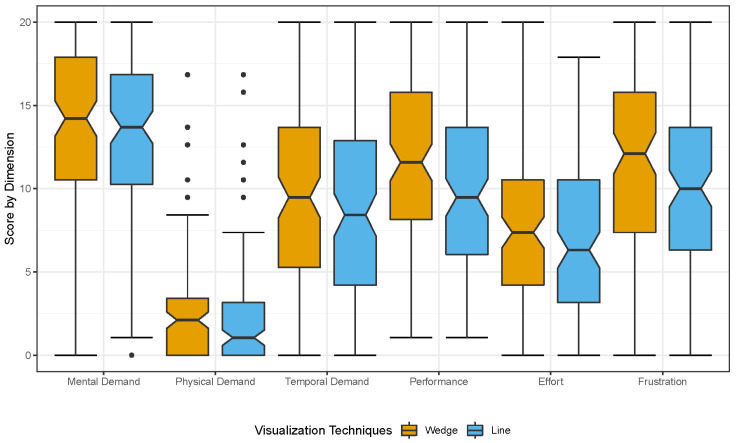
Comparison of task load dimensions for the two different visualization techniques: *Wedge* and *Line*.

**Table 1 sensors-22-00755-t001:** Pairwise comparisons of accuracy for the visualization techniques: *Wedge*, *Halo*, and *Line*.

Comparison	W	Z	*p*	r
Wedge vs. Halo	33	1.39	0.563	0.30
Wedge vs. Line	2	−2.25	0.070	0.48
Halo vs. Line	1	−2.55	0.023 *	0.54

* *p* ≤ 0.05.

**Table 2 sensors-22-00755-t002:** Pairwise comparisons of reaction times for the visualization techniques: *Wedge*; *Halo*; and *Line*.

Comparison	W	Z	*p*	r
Wedge vs. Halo	8	−2.22	0.073	0.47
Wedge vs. Line	59	2.31	0.056	0.49
Halo vs. Line	64	2.76	0.009 **	0.59

** *p* ≤ 0.01.

## Data Availability

Not applicable.

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
