# Peer review of "My Caregiver the Cobot: Comparing Visualization Techniques to Effectively Communicate Cobot Perception to People with Physical Impairments"

_sensors, 2022, doi:10.3390/s22030755_

Round 1

Reviewer 1 Report

The authors introduce their work about the comparison of three off-screen visualization techniques. The comments are as follows:

  1. communicate the cobot's perception. It is very confusing. (do you mean to exchange information or perceptional results?) There are several locations where the same expression is used in this paper.
  2. I suggest using highlighted dot to represent the objects in figure 1 for line, wedge, and halo visualization. Or, it isn't easy to express that the meaning of off-screen for the objects. Besides, the structure of the paper might be adjusted since figure 1 appeared on page2, but the descriptions were placed on page 7. It is not readable for readers.
  3. For figure 3. The projection orientation. Besides the projection orientation, do you think about the ambient light and background affections? As you know, when we use ABB cobot or the Universal cobot, it is critical to figure out these affections.
  4. The symbols and abbreviations should be clarified. For example, SD for standard deviation, chi-square test, etc... Or, it is not easy to read for ordinary readers. Same as in Tables. In addition, the chi-square symbols are not unified (see page 11, line 467. Page 12, line 475, etc.)
  5. I did not find any part to discuss recognition of an object, so please modify the claim and only focus on the comparison of the off-screen visualization techniques.

Author Response

Dear Reviewer 1,
We thank you for your thoughtful feedback. Valid issues were raised and we already included the below-outlined improvements to the paper to alleviate these concerns.

  1. The term “communicate cobot’s perception”.
    We formulated a revised version of the definition of the term in line 40: "We define cobot perception as the sensory information the cobot acquires to computationally understand its surroundings, such as the detection and identification of objects of interest in its physical vicinity. In our work, we communicate this sensory information the cobot acquires using three visualization techniques."
  2. Using a highlighted dot to represent the objects in figure 1 and it’s appearance in the article.
    The first time we refer to Figure 1 is in the introduction (line 91) to give the reader an idea of the different visualizations. We decided to move Figure 1 to page 3 to deduce distance. Based on the suggestions of Reviewer 2, we changed its caption to: "The compared visualization techniques to communicate cobot's perception are a) Line, b) Wedge, and c) Halo".
    Further, we added another figure in "Section 3.2.3. Selected Visualization Techniques" to describe the visualizations in more detail and highlight the different parts to illustrate the difference between visualizations for off- and on-screen objects.
    The reference to Figure 1 in line 176 will be changed to Figure 2a because it is a more accurate picture to describe a breakfast scenario.
  3. Consideration of ambient light and background affections for the physical setup.
    We added a paragraph in the limitations section to highlight the need to consider this when using Spatial Augmented Reality with industrial cobots: “Besides the projection orientation, special considerations regarding ambient light and background affections have to be taken into account when using SAR. Our investigation neglected these external factors as they are present independent of the projection orientation. Nevertheless, more research is required to understand their influence better.”.
  4. The symbols and abbreviations should be clarified.
    To clarify symbols and abbreviations, we added a paragraph into “Section 4.5. Results” for “SD: standard deviation and χ2(2): chi-squared with two degrees of freedom”.
    Thanks for noticing. We unified the chi-square symbols to the correct one
  5. Recognition of an object
    We are sorry for the confusion concerning object recognition of the cobot. We think that the reviewer refers to lines 45 and 140, in which we mention object recognition as a potential source of errors in autonomous systems. To avoid confusion, we changed the wording here to object perception, as follows:
    Line 45: “Users would benefit tremendously from receiving information and eventually understanding a cobot’s spatial perception as cobot perception failures, e. g., errors in computer vision and object perception, can occur and are otherwise both difficult to predict and understand [16,17].”;
    Line 140: “Especially the non-perception of objects can have fatal consequences as often demonstrated in autonomous driving4”.

In addition to these changes above, we corrected some minor English language and style issues.
We appreciate the valuable feedback.

Reviewer 2 Report

The paper reads very well. The manuscript is well written and organized. The methods and analysis are sound without major problems. Some minor improvements to be made:

- Page 4

Task selection is critical. More rationale behind selecting assistance during breakfast is needed beyond being a routine of everyday tasks.

- Page 5

The experimental task should be specified in more detail. 

Line 208: "At some point" won't do. Was there randomization in time? Please report the exact procedure with a specific timeline.

- Page 5

Instead of writing "12 and 116," the N (number of participants) should be written with the respective experiment. Specifically, "after this, we run a second experiment with (how many?) participants without impairments."

- Page 14

Trust in automation is a well-studied construct. The analysis seems shallow. I recommend enticing the findings with the current literature.

- Figure 1.

The explanation should be located in the main text and not in the Figure label.

Author Response

Dear Reviewer 2,
We thank you for your thoughtful feedback. We appreciate that you find the paper “well written and organized” and “the methods and analysis are sound without major problems”.

Valid issues were raised and we already included the below-outlined improvements to the paper to alleviate these concerns.

  1. More rationale behind selecting assistance during breakfast.
    We added more details concerning our decision in lines 176-178 as follows: “Our previous work  — an ethnographic study to conclude recommendation for the development of a robotic drinking and eating aid — has shown a clear need of an (semi-) autonomous assistive technology to support during breakfast situations [7]. Hence, we focus on a breakfast situation at a kitchen table (see Figure 2a).”.
    In addition, we changed the reference from Figure 1 to Figure 2a because it is a more accurate picture to describe a breakfast scenario.
  2. The experimental task should be specified in more detail.
    In the approach section (section 3.1) of our submitted version, we aim to give only a short overview (section 3.1.) about the design of both experiments. Therefore, we added the reference to highlight this to the reader: “(see Section 4.4. Procedure)" and "(see Section 5.4. Procedure)".
    In line 208, we changed "At some point" to: "After a randomized time in an interval of 5 to 15 seconds in experiment I (see Section 4.4 Procedure) and an interval between 3 and 15 seconds in experiment II (see Section 5.4 Procedure), the cobot would no longer detect a seemingly random object, and thereby the visualization highlight would vanish.".
  3. Writing the number of participants.
    In line 183, we changed the sentence to: “To that end, we conducted two independent remote studies with 12 participants in the first and 116 participants in a second study, analyzing a total of three different visualizations.”
    In line 187, we added the number of participants and changed the sentence to: “After this, we conducted a second experiment (see Section 5 Experiment II: Validation Study) with 116 participants without impairments to verify our findings.” To be more consistent, we changed in line 102 the term “participants without any restrictions” to “ participants without impairments”, following the advice of the SIGACCESS community.
  4. Trust in automation.
    We added to our limitations section of experiment I “Section 4.6.4. Limitations”: “In our experiment, we did not measure trust with standardized questionnaires. Nonetheless, during our interviews, participants reported insight into how they would trust the cobot in this scenario. Hence, we did not address trust in-depth, i.e., with standardized measures, and, thereby, our results can only be the foundation for further hypotheses and research concerning trust in cobots.
  5. Caption of Figure 1.
    We reduced the caption to: “The compared visualization techniques to communicate cobot's perception are a) Line, b) Wedge, and c) Halo”.

In addition to these changes above, we corrected some minor English language and style issues.
We appreciate the valuable feedback.